# Asymptotics of Subsampling for Generalized Linear Regression Models under Unbounded Design

**DOI:** 10.3390/e25010084

**Published:** 2022-12-31

**Authors:** Guangqiang Teng, Boping Tian, Yuanyuan Zhang, Sheng Fu

**Affiliations:** 1School of Mathematics, Harbin Institute of Technology, Harbin 150001, China; 2School of Mathematical Sciences, Soochow University, Suzhou 215006, China; 3Department of Industrial Systems Engineering & Management, National University of Singapore, 21 Lowr Kent Ridge Road, Singapore 119077, Singapore

**Keywords:** generalized linear models, massive data, nonnatural links, unbounded covariates, unconditional subsampling estimator

## Abstract

The optimal subsampling is an statistical methodology for generalized linear models (GLMs) to make inference quickly about parameter estimation in massive data regression. Existing literature only considers bounded covariates. In this paper, the asymptotic normality of the subsampling M-estimator based on the Fisher information matrix is obtained. Then, we study the asymptotic properties of subsampling estimators of unbounded GLMs with nonnatural links, including conditional asymptotic properties and unconditional asymptotic properties.

## 1. Introduction

In recent years, the amount of information that people need to process is increasing dramatically. It is of great challenge to directly process massive data for statistical analysis. The divide-and-conquer strategy can mitigate the challenge of directly processing such big data [1], but it still consumes considerable computing resources. As a cheaper alternative in computing, subsampling gains its value in the case of limited computing resources.

To reduce the burden on the machine, the subsampling strategy based on big data has been given more attention in recent years. Ref. [2] proposes simple necessary and sufficient conditions for a convolved subsampling estimator to produce a normal limit that matches the target of bootstrap estimation; Ref. [3] provides an optimally distributed subsampling for maximum quasi-likelihood estimators with massive data; Ref. [4] studies some adaptive optimal subsampling algorithms; and Ref. [5] describes a subdata selection method based on leverage scores which conduct the linear model selection on a small subdata set.

GLM is a kind of statistical model with a wide range of applications such as [6,7,8]. Many subsampling studies are based on GLMs such as [3,9,10]. However, the covariates of the subsampled GLMs in the literature are bounded. When dealing with some big data problems, the size of covariate is not strictly bounded, such as the number of clicks on a web page, which can grow infinitely. This requires the extension of existing theories to the unbounded design. To fill this gap, this paper aims to study asymptotic properties of the subsampled GLMs with unbounded covariates based on empirical process and martingale technology.

Our three contributions are shown as follows: (1) we describe the asymptotic property of subsampled M-estimator using Fisher information matrix; (2) we give the conditional consistency and asymptotic normality of unbounded GLMs subsampling estimator; (3) we provide the unconditional consistency and asymptotic normality of unbounded GLMs subsampling estimator.

The rest of the paper is organized as follows. Section 2 introduces the basic concepts in GLMs and subsampling M-estimation problem. Section 3 presents the asymptotical properties for unbounded GLMs subsampling estimators. Section 4 gives the conclusion and discussion, as well as future research directions. All the technical proofs are collected in the Appendix A.

## 2. Preliminaries

This section introduces the subsampling M-estimation problem and GLMs.

### 2.1. Subsampling M-Estimation

Let {l(β;Z)∈R|Z∈Z} be a set of loss functions with a finite dimensional convex set $\beta \in \Theta \subset R^{p}$, and U={1,2,…,N} be the index of the full large dataset with σ-algebra $F_{N}=\sigma \left(Z_{1},\ldots ,Z_{N}\right)$, where for each i∈U, the random data point $Z_{i}\in Z$ (some probability space) is observed. The empirical risk $L_{N}:\Theta \to R$ is given by $L_{N}\left(\beta \right)=\frac{1}{N}\sum_{i\in U}l\left(\beta ;Z_{i}\right)$.

The goal is to get the solution ${\hat{\beta }}_{N}$ to minimize the risk, namely
(1)$${\hat{\beta }}_{N}=\arg\min_{\beta \in \Theta }L_{N}\left(\beta \right).$$

To solve Equation (Equation 1), we need ${\hat{\beta }}_{N}$ satisfy: $\nabla L_{N}\left(\beta \right)=\frac{1}{N}\sum_{i\in U}\nabla l\left(\beta ;Z_{i}\right)=0$, and let $\Sigma_{N}:=\nabla^{2}L_{N}\left({\hat{\beta }}_{N}\right)$. This is an M-estimation problem; see [11]. For fast solving large-scale estimation in Equation (Equation 1), we propose the subsampling M-estimation. Consider an index set $S=\{i_{1},i_{2},\ldots ,i_{n}\}$ with replacement from *U* according to the sampling probability ${\left\{\pi_{i}\right\}}_{i=1}^{N}$ such that $\sum_{i=1}^{N}\pi_{i}=1.$ The subsampling M-estimation problem is to obtain the solution ${\hat{\beta }}_{n}$ satisfying
$$\nabla L_{n}\left(\beta \right)=0\;\;\mathrm{with}\;\;L_{n}\left(\beta \right)=\frac{1}{Nn}\sum_{i\in S}\frac{1}{\pi_{i}^{*}}l\left(\beta ;Z_{i}^{*}\right),$$
where $Z_{i}^{*}$ is the *i*-th time subsample with replacement and $\pi_{i}^{*}$ is the subsampling probability of $Z_{i}^{*}$. For example, if $Z_{1}^{*}=Z_{1}$, then $\pi_{1}^{*}=\pi_{1}$; if $Z_{2}^{*}=Z_{1}$, then $\pi_{2}^{*}=\pi_{1}$. Denote $a_{i}$ as the number of *i*-th subsampled data such that $\sum_{i\in U}a_{i}=n$. And $L_{n}\left(\beta \right)$ is constructed by inverse probability weighting skill such that $E\left[L_{n}\left(\beta \right)\right|F_{N}]=L_{N}\left(\beta \right)$; see [12]. Details about properties of conditional expectation are shown in [13].

### 2.2. Generalized Linear Models

Let the random variable *Y* be the distribution of the natural exponential families $P_{\alpha }$ indexed by parameter α,
(2)$$P_{\alpha }\left(dy\right)=dF_{Y}\left(y\right)=c\left(y\right)\exp\left\{y\alpha -b\left(\alpha \right)\right\}\nu \left(dy\right),c\left(y\right)>0,$$
where α is often referred to as the canonical parameter belonging to its natural space
Λ={α:∫c(y)exp{yα}ν(dy)<∞}.
ν(·) is the Lebesgue measure for continuous distributions (Normal, Gamma) or counting measure for discrete distributions (binomial, Poisson, negative binomial). The c(y) is free of α.

Let ${\left\{\left(Y_{i},X_{i}\right)\right\}}_{i=1}^{N}$ be *N* independent sample data pairs. Here the $X_{i}\in R^{p}$ is covariates and we assume that the response $Y_{i}$ follows the distribution of the natural exponential families with the parameter $\alpha_{i}\in \Lambda$. The covariates $X_{i}:={\left(x_{i1},\ldots ,x_{ip}\right)}^{T},\left(i=1,2,\ldots ,N\right)$ are supposed to be deterministic.

The conditional expectation of $Y_{i}$ for a given $X_{i}$ is defined as a function of $\beta^{T}X_{i}$ after a transformation by a link function $\alpha_{i}=\psi \left(\beta^{T}X_{i}\right)$. The mean value denoted as $\mu_{i}:=E\left(Y_{i}\right)$ is mostly considered for regression.

If $\alpha_{i}=\beta^{T}X_{i}$ then we call that $\psi \left(\beta^{T}X_{i}\right)=\beta^{T}X_{i}$ is canonical (or natural) link function, and corresponding model is canonical (or natural) GLMs; see Page 32 in [14]. Sometimes the assumption $\alpha_{i}=\beta^{T}X_{i}$ is somewhat strong and not very suitable in practice, while nonnatural link GLMs allow more flexible choices for the link function. We can further assume that $\alpha_{i}$ and $\beta^{T}X_{i}$ can be related by a nonnatural link function $\alpha_{i}=\psi \left(\beta^{T}X_{i}\right)$.

Let $f_{\beta }\left(Y_{i}|X_{i}\right)$ be the joint density function of the i.i.d. data ${\left\{\left(Y_{i},X_{i}\right)\right\}}_{i=1}^{N}$ from the exponential family with a link function ψ(·). Then the nonnatural GLMs [15] is defined by
(3)$$Y_{i}|X_{i}∼f_{\beta }\left(Y_{i}|X_{i}\right)=c\left(Y_{i}\right)\exp\left\{Y_{i}\psi \left(\beta^{T}X_{i}\right)-b\left(\psi \left(\beta^{T}X_{i}\right)\right)\right\},\;i=1,2,\ldots ,N.$$

Here is a classic result for the exponential family (Equation 3),
(4)$$E\left(Y_{i}|X_{i}\right):=\mu_{i}=\dot{b}\left(\alpha_{i}\right)=\dot{b}\left(\psi \left(\beta^{T}X_{i}\right)\right)\;and\;\mathrm{Var}\left(Y_{i}|X_{i}\right):=\mathrm{Var}\left(Y_{i}\right)=\ddot{b}\left(\alpha_{i}\right),$$
where i=1,2,…,N; see P280 in [16].

## 3. Main Results

### 3.1. Subsampling M-Estimation Problem

In this part we first look at the term $\Sigma_{N}^{-1}\nabla L_{n}\left({\hat{\beta }}_{N}\right)$. Define an independent random vector sequence ${\left\{\zeta_{j}\right\}}_{j=1}^{N}$ and the subsampled ${\left\{\zeta_{j}^{*}\right\}}_{j=1}^{n}$, such that each vector ζ takes the value among ${\left\{\frac{1}{N\pi_{i}}\Sigma_{N}^{-1}\nabla l\left({\hat{\beta }}_{N};Z_{i}\right)\right\}}_{i=1}^{N},$ and let
$$V_{M}\left({\hat{\beta }}_{N};n\right)=\frac{1}{N^{2}n}\Sigma_{N}^{-1}\left[\sum_{i\in U}\frac{1}{\pi_{i}}\nabla l\left({\hat{\beta }}_{N};Z_{i}\right)\nabla^{T}l\left({\hat{\beta }}_{N};Z_{i}\right)\right]\Sigma_{N}^{-1}.$$

From the definition of $\nabla L_{N}\left(\beta \right)$, we have $E\left(\zeta |F_{N}\right)=\Sigma^{-1}\nabla L_{N}\left({\hat{\beta }}_{N}\right)=0$ and $\mathrm{Var}(\zeta |F_{N})$$=nV_{M}\left({\hat{\beta }}_{N};n\right).$ Then we have the asymptotic property of subsampled M-estimator.

**Theorem** **1.**
*Suppose that the risk function $L_{N}\left(\beta \right)$ is twice differentiable and λ-strongly convex over *Θ*, that is, for β∈Θ, $\nabla^{2}L_{N}\left(\beta \right)\geq \lambda I$, where ≥ denotes the semidefinite positive ordering; and the sampling-based moment condition,*

$$\begin{array}{c} \frac{1}{N^{4}}\sum_{i=1}^{N}\frac{1}{\pi_{i}^{3}}{∥\nabla l({\hat{\beta }}_{N};Z_{i})∥}^{4}=O_{P}\left(1\right). \end{array}$$

*Then we can obtain: As n→∞, conditioning on $F_{N}$,*

(5)
$$V_{M}{\left({\hat{\beta }}_{N};n\right)}^{-\frac{1}{2}}\left({\hat{\beta }}_{n}-{\hat{\beta }}_{N}\right)\overset{d}{\to }N\left(0,I_{p}\right),$$

*where $\overset{d}{\to }$ means convergence in distribution.*


Theorem 1 reveals that the subsampling M-estimation scheme is theoretically feasible under mild conditions. In addition, the existence of the estimator is given by the Fisher information matrix.

### 3.2. Conditional Asymptotic Properties of Subsampled GLMs with Unbounded Covariates

The exponential family is very versatile for containing many common light-tail distributions such as binomial, Poisson, negative binomial, normal and Gamma. Along with their attendant convexity properties which leads to finite variance property for log-density, they can serve for a large amount of popular and effective statistical models. It is precisely because of the commonality of these distributions so that we study the subsampling problem for GLMs.

From the loss function introduced in Section 2.1, we set $l\left(\beta ;Z_{i}\right):=-\log f_{\beta }\left(Y_{i}|X_{i}\right)$ where $f_{\beta }\left(Y_{i}|X_{i}\right)$ is defined by Equation (Equation 2), then the problem solving the minimum of the loss function is equivalent to solve the maximum of the likelihood function. For simplicity, we assume that c(y)=1, then
$$\nabla l\left(\beta ;Z_{i}\right):=-\frac{\partial \log f_{\beta }\left(Y_{i}|X_{i}\right)}{\partial \beta }=-\left[Y_{i}-\dot{b}\left(\psi \left(\beta^{T}X_{i}\right)\right)\right]\dot{\psi }\left(\beta^{T}X_{i}\right)X_{i}$$
with the nonnatural link function $\alpha_{i}=\psi \left(\beta^{T}X_{i}\right)$. We also use this idea in Section 3.3.

More generally, we consider a wider class saying quasi-GLMs, rather than GLMs, which assumes that Equation (Equation 4) holds for a certain function μ(·). Strong consistency and asymptotic normality of quasi maximum likelihood estimate in GLMs with bounded covariates are proved in [17]. For unbounded covariates, adopting the subsampled estimation of GLMs in [9], we calculate the inverse probability weighted estimator of β by solving the estimating equation based on the subsampled index set *S*,
$$-\frac{1}{Nn}\sum_{i\in S}\frac{1}{\pi_{i}^{*}}\left[Y_{i}^{*}-\mu \left(\psi \left(\beta^{T}X_{i}^{*}\right)\right)\right]\dot{\psi }\left(\beta^{T}X_{i}^{*}\right)X_{i}^{*}=0.$$
where ${\left\{\left(Y_{i}^{*},X_{i}^{*}\right)\right\}}_{i\in S}$ is subsampled data. Equivalently, we have
(6)$$s_{n}\left(\beta \right)=\sum_{i\in S}\frac{1}{\pi_{i}^{*}}\left[Y_{i}^{*}-\mu \left(\psi \left(\beta^{T}X_{i}^{*}\right)\right)\right]\dot{\psi }\left(\beta^{T}X_{i}^{*}\right)X_{i}^{*}=0.$$

Equation (Equation 6) is called quasi-GLMs since Equation (Equation 4) is given instead of the distribution function.

Let ${\hat{\beta }}_{n}$ be the estimator of the real parameter $\beta_{0}$ in subsampled quasi-GLMs and ${\hat{\beta }}_{N}$ be the estimator of $\beta_{0}$ in quasi-GLMs with full data. For the unbounded quasi-GLMs with full data, ${\hat{\beta }}_{N}$ is asymptotic unbiased with respect to $\beta_{0}$; see [18]. Next, we focus on the asymptotical properties of ${\hat{\beta }}_{n}$, as shown in the following theorems.

**Theorem** **2.**
*Let ${\left\{\left(Y_{i}^{*},X_{i}^{*}\right)\right\}}_{i\in S}$ be subsampled from i.i.d. full data ${\left\{\left(Y_{i},X_{i}\right)\right\}}_{i\in U}$. Consider the Equation (Equation 4) and (Equation 6) where ψ(·) is three times continuously differentiable whose every derivative is bounded, and b(·) is twice continuously differentiable whose every derivative is also bounded. Assume that:*
*(A.1)* 
*The range of the unknown parameter β is an open subset of $R^{p}$.*
*(A.2)* 
*For any i∈S, $E\sup_{\beta \in \Theta }\left[\frac{1}{\pi_{i}^{*}}\left|Y_{i}^{*}-\mu \left(\psi \left(\beta^{T}X_{i}^{*}\right)\right)\right||F_{N}\right]=O\left(1\right)$.*
*(A.3)* 
*For any β∈Θ and i∈S, $0<\inf_{i}\varphi \left(\beta^{T}X_{i}^{*}\right)\leq \sup_{i}\varphi \left(\beta^{T}X_{i}^{*}\right)<\infty$, where $\varphi \left(t\right)={\left[\dot{\psi }\left(t\right)\right]}^{2}\ddot{b}\left(\psi \left(t\right)\right)$.*
*(A.4)* 
*For any $\beta_{1}\in \Theta$ and $\beta_{2}\in \Theta$, there exists a function $|m(X_{i}^{*})|<\infty$ such that*

$$|\varphi \left(\beta_{1}^{T}X_{i}^{*}\right)-\varphi \left(\beta_{2}^{T}X_{i}^{*}\right)|\leq |m\left(X_{i}^{*}\right)\left|\right|\beta_{1}^{T}X_{i}^{*}-\beta_{2}^{T}X_{i}^{*}|.$$

*(A.5)* 
*When $n\to \infty ,\max_{i\in S}X_{i}^{*T}{\left(X^{*}X^{*T}\right)}^{-1}X_{i}^{*}=O\left(n^{-1}\right)$ and $\lambda_{\min}\left[X^{*}X^{*T}\right]\to \infty$, where $X^{*}=\left(X_{1}^{*},...,X_{n}^{*}\right)$ and $\lambda_{\min}\left[A\right]$ is the smallest eigenvalue of the matrix A.*
*(A.6)* 
*$\min_{i=1,...,N}\left(N\pi_{i}\right)=O\left(1\right)$, $\max_{i=1,...,N}\left(N\pi_{i}\right)=O\left(1\right)$.*


*Then ${\hat{\beta }}_{n}$ is consistent with ${\hat{\beta }}_{N}$, i.e.,*

$$∥{\hat{\beta }}_{n}-{\hat{\beta }}_{N}∥=o_{P|F_{N}}\left(1\right)$$

*where $o_{P|F_{N}}\left(1\right)$ means o(1) conditioning on $F_{N}$ in probability.*


**Theorem** **3.**
*Under the conditions in Theorem 2, as N→∞ and n→∞, conditional on $F_{N}$ in probability,*

$$\sqrt{n}\left({\hat{\beta }}_{n}-{\hat{\beta }}_{N}\right)\to N\left(0,V_{s}\right),$$

*in distribution, where*

$$\begin{array}{cc} & V_{s}=\Sigma_{N}^{-1}V_{N}\Sigma_{N}^{-1},\\ & \Sigma_{N}=\sum_{i\in U}a_{i}\left[Y_{i}-\dot{b}\left(\psi \left({\hat{\beta }}_{N}^{T}X_{i}\right)\right)\right]\ddot{\psi }\left({\hat{\beta }}_{N}^{T}X_{i}\right)X_{i}X_{i}^{T}-\sum_{i\in U}a_{i}\ddot{b}\left(\psi \left({\hat{\beta }}_{N}^{T}X_{i}\right)\right)\dot{\psi }{\left({\hat{\beta }}_{N}^{T}X_{i}\right)}^{2}X_{i}X_{i}^{T},\\ & V_{N}=\sum_{i\in U}\frac{a_{i}}{\pi_{i}}{\left[Y_{i}-\dot{b}\left(\psi \left({\hat{\beta }}_{N}^{T}X_{i}\right)\right)\right]}^{2}\dot{\psi }{\left({\hat{\beta }}_{N}^{T}X_{i}\right)}^{2}X_{i}X_{i}^{T}. \end{array}$$



In this part, we complete the asymptotic properties without the moment condition of the covariates ${\left\{X_{i}\right\}}_{i=1}^{N}$ which is used in [9], and that means $X_{i}$’s are unbounded. Here we only provide the theoretical asymptotic results. Furthermore, the subsampling probability can be derived by A-optimal criterion like [10].

### 3.3. Unconditional Asymptotic Properties of Subsampled GLMs with Unbounded Covariates

In real engineering, the measurement of some response variable data is very expensive, such as superconductor data, deep space exploration data, etc. The accuracy of estimating the target parameters under measurement constraints of responses is a very important issue. Ref. [19] completed the unconditional asymptotic properties of parameter estimation in bounded GLMs with canonical link. But the unbounded GLMs with nonnatural link situation has not been discussed yet.

In this section, we continue to use the notations of Section 3.2. Through the theory of empirical process [11], we obtain the unconditional consistency of ${\hat{\beta }}_{n}$ in the following theorem.

**Theorem** **4.**
*(Unconditional subsampled consistency) Assume the conditions:*
*(B.1)* 
*$\lambda_{\min}\left(EXX^{T}\right)>0$ where X is the unbounded covariate of GLMs.*
*(B.2)* 
*For $\forall u_{1},u_{2}\in \left[0,1\right]$,*

$$\underset{\beta \in \Theta \backslash \{\beta_{0}\}}{\inf}\frac{E\{\ddot{b}\left({\tilde{\psi }}_{u_{1}}\right)\dot{\psi }{\left[\left(1-u_{2}\right)\left(\beta_{0}^{T}X\right)+u_{2}\left(\beta^{T}X\right)\right]}^{2}{\left(\beta^{T}X-\beta_{0}^{T}X\right)}^{2}\}}{E{\left(\beta^{T}X-\beta_{0}^{T}X\right)}^{2}}\geq C_{1}>0,$$

*where ${\tilde{\psi }}_{u_{1}}=\left(1-u_{1}\right)\psi \left(\beta_{0}^{T}X\right)+u_{1}\psi \left(\beta^{T}X\right)$ and $\ddot{b}\left(\cdot \right)$ is the second derivative with respect to β.*
*(B.3)* 

$$E_{\beta_{0}}\underset{\beta \in \Theta }{\sup}[|Y-\dot{b}\left(\psi \left(\beta^{T}X\right)\right){\left|\cdot ||X|\right|}^{2}]<\infty ,$$

*where $\dot{b}\left(\cdot \right)$ is the first derivative with respect to β.*
*(B.4)* 
*ψ(·) in (Equation 3) is twice continuously differentiable and its every derivative has a positive minimum.*
*(B.5)* 
*b(·) in (Equation 3) is twice continuously differentiable and its every derivative has a positive minimum.*


*Then $∥{\hat{\beta }}_{n}-\beta_{0}∥=o_{P}\left(1\right)$.*


Theorem 4 directly obtains the unconditional consistency of the subsampling estimator with respect to the true parameters under the unbounded assumption.

To prove the asymptotic normality of ${\hat{\beta }}_{n}$ with respect to $\beta_{0}$, we briefly review the subsampled score function in Section 3.2
$$s_{n}\left(\beta \right)=\sum_{i\in S}\frac{1}{\pi_{i}^{*}}\left[Y_{i}^{*}-\mu \left(\psi \left(\beta^{T}X_{i}^{*}\right)\right)\right]\dot{\psi }\left(\beta^{T}X_{i}^{*}\right)X_{i}^{*}:=\sum_{i\in S}\frac{1}{\pi_{i}^{*}}\phi_{\beta }\left(X_{i}^{*},Y_{i}^{*}\right).$$

Next we will apply a multivariate martingale central limit theorem (Lemma 4 in [19]), which is the extension of Theorem A.1 in [20], to show the asymptotic normality of ${\hat{\beta }}_{n}$. Let ${\left\{F_{N,i}\right\}}_{i=1}^{n}$ be a filtration adaptive to the sampling: $F_{N,0}=\sigma \left(X_{1}^{N},Y_{1}^{N}\right);F_{N,1}=\sigma \left(X_{1}^{N},Y_{1}^{N}\right)\vee \sigma \left({*}_{1}\right);\ldots ;F_{N,i}=\sigma \left(X_{1}^{N},Y_{1}^{N}\right)\vee \sigma \left({*}_{1}\right)\vee \ldots \vee \sigma \left({*}_{i}\right);\ldots$, where $\sigma ({*}_{i})$ is the σ-algebra generated by *i*th sampling step. The subsample of size *n* is assumed to increase with *N*. By the filtration, we define the martingale
$$\bar{M}:=\sum_{i=1}^{n}{\bar{M}}_{i}:=\sum_{i=1}^{n}\left[\frac{1}{\pi_{i}^{*}}\phi_{\beta }\left(X_{i}^{*},Y_{i}^{*}\right)-\sum_{j=1}^{N}\phi_{\beta }\left(X_{j},Y_{j}\right)\right],$$
where ${\left\{{\bar{M}}_{i}\right\}}_{i=1}^{n}$ is a martingale difference sequence adapted to ${\left\{F_{N,i}\right\}}_{i=1}^{n}$. In addition, define $Q:=n\sum_{j=1}^{N}\phi_{\beta }\left(X_{j},Y_{j}\right)$; $T:=s_{n}\left(\beta \right)=\bar{M}+Q$; $\xi_{Ni}:={\mathrm{Var}}^{-1/2}\left(T\right){\bar{M}}_{i}$ and $B_{N}:={\mathrm{Var}}^{-1/2}\left(T\right)\mathrm{Var}\left(\bar{M}\right){\mathrm{Var}}^{-1/2}\left(T\right)$, where matrix $A^{1/2}$ is the symmetric square root of A, i.e., $A={\left(A^{1/2}\right)}^{2}$, and $A^{-1/2}={\left(A^{1/2}\right)}^{-1}={\left(A^{-1}\right)}^{1/2}$. $B_{N}$ is the variance of ${\mathrm{Var}}^{-1/2}\left(T\right)\bar{M}$.

The following theorem shows the asymptotic normality of the estimator ${\hat{\beta }}_{n}$.

**Theorem** **5.**
*Assume the conditions,*
*(C.1)* 

$$\Phi =E\left(\nabla s_{n}\left(\beta \right)\right)=E\left[-\sum_{i\in S}\frac{1}{\pi_{i}^{*}}\dot{\mu }\left(\psi \left(\beta^{T}X_{i}^{*}\right)\right){\left[\dot{\psi }\left(\beta^{T}X_{i}^{*}\right)\right]}^{2}X_{i}^{*}X_{i}^{*T}\right]$$

*is finite and nonsingular.*
*(C.2)* 
*$E\left\{{\left[\sum_{i\in U}\frac{a_{i}}{\pi_{i}}\dot{\mu }\left(\psi \left(\beta^{T}X_{i}\right)\right){\left[\dot{\psi }\left(\beta^{T}X_{i}\right)\right]}^{2}X_{ik}X_{ij}\right]}^{2}\right\}=o_{P}\left(1\right)$, for 1≤k,j≤p,*
*where $X_{ik}$ means k-th element of vector $X_{i}$ and $X_{ij}$ means j-th element of vector $X_{i}$.*
*(C.3)* 
*ψ(x) is three-times continuously differentiable for every x with its domain.*
*(C.4)* 
*For any i∈S, $\left|\right|{\ddot{\phi }}_{\beta }\left(X_{i}^{*},Y_{i}^{*}\right)\left|\right|<\infty$.*
*(C.5)* 
*$\min_{i=1,...,N}\left(N\pi_{i}\right)=\max_{i=1,...,N}\left(N\pi_{i}\right)=O\left(1\right)$ and n/N=o(1).*
*(C.6)* 
*$\lim_{N\to \infty }\sum_{i=1}^{n}E[||\xi_{Ni}{\left|\right|}^{4}]=0$,*
*(C.7)* 

$\lim_{N\to \infty }E\left[{\left|\left|\sum_{i=1}^{n}E\left[\xi_{Ni}\xi_{Ni}^{T}|F_{N,i-1}\right]-B_{N}\right|\right|}^{2}\right]=0.$



*Then*

$$\mathrm{Var}{\left(T\right)}^{-1/2}\Phi \left({\hat{\beta }}_{n}-\beta_{0}\right)\overset{d}{\to }N\left(0,I_{p}\right).$$



Here, we establish the unconditional asymptotic properties of subsampling estimator for unbounded GLMs. The condition n/N=o(1) ensures that small-scale subsamples also have expected performance, which greatly release the computational cost. We also present the theoretical asymptotic results, which leads to the subsampling probability using the A-optimal criterion in [10].

## 4. Conclusions and Future Work

In this paper, we derive the asymptotic normality of the subsampling M-estimator by Fisher information. In the unbounded GLMs with nonnatural link function, we separately obtain the conditional and unconditional asymptotic properties of subsampling estimator.

For future study, it is meaningful to apply the sub-Weibull concentration inequalities in [21] to make nonasymptotic inference. The importance sampling is not ideal, since it tends to assign high sampling probability to the observed samples. Hence, effective subsampling methods are considered for GLMs, such as Markov subsampling in [22]. Moreover, high-dimensional methods in [23,24] for subsampling need further studies. 

## Data Availability

Not applicable.

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
