# Peer review of "Asymptotics of Subsampling for Generalized Linear Regression Models under Unbounded Design"

_entropy, 2022, doi:10.3390/e25010084_

Round 1
Reviewer 1 Report
This paper establishes the conditional and unconditional asymptotic normalities of a subsample-based estimator for a generalized linear regression coefficient without requiring the covaraite to be bounded. It may contains something new in theory, however the presentation is not good. Also there are many typos or grammatical errors. Below are some of them.
1. Condition (A6) of Theorem 2 implies that pi_i are all equal to each other, which is absurd.
2. In the second paragraph of the introduction, the reference [10] namely Jordan, Lee and Yang (2018) is irrelated to subsampling.
3. In the beginning of 2.1, what do you mean by "a collection convex loss function"?
4. In the paragraph below equation (2), the Inverted triangle is used before defined. The two paragraphes above and below equation (2) do not read smooth.
5. The sentence above equation (5) is a bit repetitive. Also the symbol of convergence in distribution is used before defined.
Reviewer 2 Report
the paper entailed " Asymptotic of Subsampling for Generalized Linear Regression Models under Unbounded Design " the authors present
The optimal subsampling is a statistical methodology for generalized linear models 2 (GLMs) to make inference quickly about parameter estimation in massive data regression. Existing 3 literature only considers bounded covariates. In this paper, we obtain the asymptotic normality 4 of the subsampling M-estimator based on the Fisher information matrix. Then, we study the 5 asymptotic properties of subsampling estimators of unbounded GLMs with no natural links, 6 including conditional asymptotic properties and unconditional asymptotic properties
the paper well written no Comments about it except the language need to review with native speaker.